# Loss of heterozygosity results in rapid but variable genome homogenization across yeast genetic backgrounds

Abhishek Dutta[1], Fabien Dutreux[1], Joseph Schacherer[1,2]*

[1]Université de Strasbourg, CNRS, GMGM UMR 7156, Strasbourg, France; [2]Institut Universitaire de France (IUF), Paris, France

**Abstract** The dynamics and diversity of the appearance of genetic variants play an essential role in the evolution of the genome and the shaping of biodiversity. Recent population-wide genome sequencing surveys have highlighted the importance of loss of heterozygosity (LOH) events and have shown that they are a neglected part of the genetic diversity landscape. To assess the extent, variability, and spectrum, we explored the accumulation of LOH events in 169 heterozygous diploid *Saccharomyces cerevisiae* mutation accumulation lines across nine genetic backgrounds. In total, we detected a large set of 22,828 LOH events across distinct genetic backgrounds with a heterozygous level ranging from 0.1% to 1%. LOH events are very frequent with a rate consistently much higher than the mutation rate, showing their importance for genome evolution. We observed that the interstitial LOH (I-LOH) events, resulting in internal short LOH tracts, were much frequent (n = 19,660) than the terminal LOH (T-LOH) events, that is, tracts extending to the end of the chromosome (n = 3168). However, the spectrum, the rate, and the fraction of the genome under LOH vary across genetic backgrounds. Interestingly, we observed that the more the ancestors were heterozygous, the more they accumulated T-LOH events. In addition, frequent short I-LOH tracts are a signature of the lines derived from hybrids with low spore fertility. Finally, we found lines showing almost complete homozygotization during vegetative progression. Overall, our results highlight that the variable dynamics of the LOH accumulation across distinct genetic backgrounds might lead to rapid differential genome evolution during vegetative growth.

*For correspondence:
schacherer@unistra.fr

## Introduction

The appearance of genetic variants is a key point in the evolution of genomes as well as in the adaptation of a species to environmental changes. Genomic variability is considered to result from recombination events as well as the generation of new mutations. In a population, the genetic diversity is in part the result of multiple processes leading to single-base mutations and large alterations such as deletions, duplications, translocations, and changes of the ploidy level. The rate of these events is low in most organisms but it has been argued that the rates have been optimized by natural selection to ensure survival (*Gerrish et al., 2007*; *Keightley et al., 2009*; *Lynch et al., 2008*; *Lynch et al., 2016*; *Ossowski et al., 2010*; *Lee et al., 2012*; *Zack et al., 2013*). At a population level, this genetic diversity lays the foundations for phenotypic diversity.

The ability to systematically sequence the genomes of large populations has provided a deeper insight into all these events at a species-wide level. The diploid organisms (or those with a higher ploidy) are usually heterozygous for many of these genetic variants across the genome. Nevertheless, a large number of studies have pointed out that regions of the genome can frequently become homozygous for these polymorphisms during mitotic divisions (*Zack et al., 2013*; *Holland and Cleveland, 2009*; *Smukowski Heil et al., 2017*; *Peter et al., 2018*; *Ropars et al., 2018*; *Lancaster et al., 2019*; *Gounot et al., 2020*; *Eberlein et al., 2021*; *Nichols et al., 2020*). These loss

of heterozygosity (LOH) events result from the transfer of information from one homologous chromosome to the other, primarily a consequence of mitotic recombination, among other mechanisms (*Sui et al., 2020*). LOH events are hence another source of genomic diversity and can contribute to the rapid onset of phenotypic diversity. In asexual populations, mechanisms driving LOH are crucial in the fixation of potential beneficial recessive mutations and counteract the Muller's ratchet, allowing them to persist (*Howe and Denver, 2008*; *Tucker et al., 2013*; *Mandegar and Otto, 2007*). LOH events have been identified as involved in fitness, adaptation, and pathogenesis in many fungi (*Forche et al., 2011*; *Wertheimer et al., 2016*; *Beekman and Ene, 2020*). In addition, LOH is also a common genetic event in cancer development (*Nichols et al., 2020*). LOH of *BRCA1/2* (protein products regulate Rad51 activity in DNA repair pathways) and *Rb1* (a human tumor suppressor gene) can lead to breast cancers and retinoblastoma, respectively (*Nichols et al., 2020*; *Maxwell et al., 2017*; *Kechin et al., 2018*; *Sorscher et al., 2020*; *Choi et al., 2002*; *Herschkowitz et al., 2008*; *Kooi et al., 2016*). Nevertheless, LOH events are genomic alterations that were overlooked in the dissection of the genotype-phenotype relationship.

The mechanisms behind LOH have been largely derived from studies based on the selection of events involving specific genes and on the use of the *Saccharomyces cerevisiae* yeast as a model organism (*St Charles and Petes, 2013*). Double-strand break (DSB) repair is a common occurrence during the cell cycle that can lead to LOH. Repair of programmed DSBs induced by Spo11 is a key point of meiosis. These DSBs are repaired via reciprocal or non-reciprocal exchanges between the homologous chromosomes, crossovers (CO), or non-crossovers (NCO). Intra-tetrad mating among haploid daughter cells can also lead to LOH events. However, inter-homolog events occur at a significantly lower frequency in mitotically dividing cells. Recombination events during mitosis are random, in contrast, it is a tightly regulated process during meiosis. Return to vegetative growth, after abortive meiosis (RTG), is another process that can generate significant LOH via mitotic repair of Spo11-induced DSBs (*Dayani et al., 2011*; *Laureau et al., 2016*). A growing body of evidence suggests that RTGs may be a common occurrence in the life cycle of yeasts (*Peter et al., 2018*; *Brion et al., 2017*; *D'Angiolo et al., 2020*). In diploids undergoing mitotic growth, non-reciprocal exchanges between homologous chromosomes give rise to LOH. It is primarily caused by gene conversions, crossing over, and break-induced replication (BIR) due to DSB repair (*Pâques and Haber, 1999*; *Symington et al., 2014*). Mitotic recombination can lead to interstitial (I-LOH) and terminal (T-LOH) LOH events. The I-LOH events are mostly caused by gene conversions involving short exchanges, ranging up to 10 kb in size (*Yim et al., 2014*). By contrast, T-LOH events result from mitotic CO and BIR mechanisms frequently spanning large chromosomal regions greater than 100 kb (*Llorente et al., 2008*; *Malkova and Ira, 2013*). BIR is a homologous recombination (HR) pathway that repairs only one-ended DSBs and suppressed at two-ended DSBs (*Pham et al., 2020*). Despite their high fidelity relative to other repair pathways, recent findings have demonstrated that HR pathways promote genome instability through an increased load of mutagenesis and chromosomal rearrangements (*Kumar et al., 2015*; *Magwene et al., 2011*; *Ruderfer et al., 2006*). Both I-LOH and T-LOH events have been implicated in cancers, with I-LOH events being observed at higher frequencies in gastric and lung cancers while T-LOH events being prevalent in retinoblastomas and colorectal cancers (*Nichols et al., 2020*; *Sui et al., 2020*; *Kumar et al., 2015*).

Recent population genomic studies have highlighted the extent of LOH events in large samples of yeast natural isolates (*Peter et al., 2018*; *Ropars et al., 2018*; *Magwene et al., 2011*). Whole-genome sequencing of 1011 *S. cerevisiae* strains coming from various ecological and geographical origins strongly supported the idea of LOH-mediated phenotypic diversification (*Peter et al., 2018*). *S. cerevisiae* is highly inbred and characterized by irregular sexual cycles (*Ruderfer et al., 2006*). Meiosis and consequently outcrossing events are infrequent, and the clonal expansion is punctuated by a broad LOH accumulation. As a consequence, a high LOH level with 25 regions covering approximately 50% of the genome on average was observed in this large panel (*Peter et al., 2018*). Nevertheless, this trend is not conserved and a highly variable proportion of the genome under LOH is observed across subpopulations, ranging from a few to 80%. As a result, the degree of heterozygosity is also diverse and can be extensive in some subpopulations (*Peter et al., 2018*). Echoing the mutation rate and its relative conservation, this observation definitely raised the question regarding the impact of genetic backgrounds on the LOH spectrum and rate.

To have a better overview of the accumulation of LOH events over time in a population, we performed a mutation accumulation (MA) experiment by using a set of different *S. cerevisiae* diploid

hybrids, designed to mimic the wide genetic diversity observed in this species. A total of 22,828 LOH events were detected across 169 MA lines propagated for at least 1769 divisions. This dataset allowed to recapitulate the LOH landscape observed in the natural population. We observed a rapid accumulation of LOH through vegetative propagation across the lines. However, we found that the LOH spectrum and rate is very variable across genetic backgrounds. Even if variable, the LOH rate is always much higher than the mutation rate, which is very conserved across the lines. Overall, our results highlight the overriding role of the LOH events in rapidly generating genetic diversity and shaping the genome as well as its broad variability across the genetic backgrounds.

## Results

### Setup and propagation of MA lines

To characterize the genome-wide landscape and dynamics of LOH accumulation, we generated 180 MA lines using nine different heterozygous *S. cerevisiae* strains (*Table 1*). The diploids were obtained by crossing haploid derivates of natural *S. cerevisiae* isolates from various ecological (e.g. tree, wine, and fruit) and geographical origins (*Peter et al., 2018*) (e.g. Africa, China, and Europe) (*Supplementary file 1*). These hybrids were designed to lead to a heterozygosity level ranging from 0.1% to 1%, covering a large genetic diversity of the *S. cerevisiae* species. In addition, the isolates were selected in order to have uniformly distributed heterozygous sites throughout the genome with a single nucleotide polymorphism (SNP) density ranging from 0.8 to 10 SNPs per kb (*Figure 1—figure supplement 1*). These nine diploids were designated as the ancestors, and 20 replicate lines were isolated from each of them to set up an MA experiment. The replicate lines were propagated purely vegetatively and subjected to single-cell bottlenecks every 48 hr for at least 75 bottlenecks on rich media (see Materials and methods). Selection is minimal, as the experiment was designed to make drift by far the main cause of genomic changes (*Halligan and Keightley, 2009*). At the end of the experiment, the genome of the 169 surviving lines was completely sequenced using a short-read Illumina strategy (see Materials and methods).

To accurately determine the total number of generations over the course of the MA experiment, we estimated the growth rate of the ancestors as well as the one of the MA lines at the end of the experiment in rich media. The growth rate of ancestral diploids was about 0.51 divisions per hour on average. No significant difference was observed between ancestors and final MA lines in terms of growth rate (*Supplementary file 2*, p>0.05 t-test). Therefore, we estimated that the MA lines have undergone an average of 24.2 divisions per bottleneck for a total of at least 1769 divisions per line.

### Overview of the LOH content in the 169 MA lines

We first sought to characterize and analyze the LOH events that occur in the 169 MA lines. For this, the events supported by at least two adjacent converted heterozygous sites were considered as under LOH and consecutive events were merged, if the disruption was not supported by two heterozygous sites (see Materials and methods). A two-site threshold was chosen to accurately define LOH

**Table 1.** Hybrid mutation accumulation (MA) lines.

| Hybrid | Cross* | Het positions | No. of sequenced lines | No. of bottlenecks | Total no. of divisions |
|--------|--------|---------------|------------------------|--------------------|------------------------|
| H1 | ABS × BKL | 9972 | 20 | 100 | 2446 |
| H2 | ABP × BFQ | 18789 | 12 | 75 | 1842 |
| H3 | BAP × BAN | 20875 | 20 | 75 | 1863 |
| H4 | BTI × ABA | 49412 | 20 | 75 | 1772 |
| H5 | ACD × AKQ | 52223 | 20 | 75 | 1777 |
| H6 | ACK × CMQ | 55570 | 20 | 100 | 2392 |
| H7 | ACG × BAK | 69456 | 19 | 75 | 1844 |
| H8 | CGD × AKE | 113241 | 19 | 75 | 1769 |
| H9 | BAM × CPG | 116475 | 19 | 100 | 2452 |

* Standardized names from *Peter et al., 2018*.

events across the nine backgrounds because of the 12-fold difference in terms of SNP density. In addition, the size of the LOH events was determined as the distance between the midpoint of the closest upstream and downstream unconverted sites.

Overall, we identified a large set of 22,828 LOH events across the 169 MA lines, with an average size of 14.1 kb (**Supplementary file 3**). It is interesting to observe that the size distribution of the detected LOH events is bimodal (**Figure 1A**). While the majority of the events (95%) are in the first category with an average size of 1.6 kb, the rest of the events (5%) in the second category have an average size of 269.2 kb. We observed that accumulated LOHs are further away from centromeres (mean distance of 272 kb) than telomeres (mean distance of 230 kb) (**Figure 1—figure supplement 2A**). These observations are consistent with the size and distribution of events reported during gene conversion, mitotic CO, or BIR events (**Mandegar and Otto, 2007**; **St Charles and Petes, 2013**; **Jeffreys and May, 2004**). In *S. cerevisiae*, LOH events can be either centromere proximal, I-LOH resulting from gene conversions, or centromere distal, T-LOH resulting from mitotic CO or BIR (**Sui et al., 2020**; **Loeillet et al., 2020**). Among the accumulated events, the I-LOHs were found to be significantly enriched representing 86.1% of the events (n = 19,660) compared to the T-LOHs with only 3,168 detected events (i.e. 13.9%) (**Figure 1B**). As expected, T-LOH events are significantly larger than I-LOH events with an average size of 55.3 and 7.5 kb, respectively (**Figure 1—figure supplement 3**, $p<10^{-3}$). And finally, we also observed a clear difference in the distribution of the T-LOH and I-LOH events across the genome (**Figure 1—figure supplement 2B–C**).

We then investigated the fraction of the genome under LOH as a consequence of the accumulated events. On average, 15.9% of the genome was under LOH but with a large variation as this fraction ranges from 0.01% to 98.6% across the lines. In fact, the 169 lines were divided into three distinct groups based on the proportion of genome under LOH (**Figure 1C**). The first two groups were represented by 113 and 43 lines with an average of 4.1% and 20.8% of genome under LOH, respectively. The last group encompassed 14 lines that exhibit more than 90% of the genome under LOH, which is significantly higher than the other two (**Figure 1—figure supplements 4** and **5**). This observation clearly shows that the genome of approximately 8% of the lines was quickly driven a quasi-homozygous state, henceforth we call them nearly homozygous lines.

All these data also allowed us to have an estimate of the LOH rates on all the MA lines. In fact, LOH rates can be defined based on the total number of sites in the genome under LOH or the total number of events per division. Using the 169 lines, we estimated the LOH rates to be $7.1 \times 10^{-5}$ per site per division and $6.5 \times 10^{-2}$ events per division on average. However, these rates include both I-LOHs and T-LOHs and we therefore estimated their respective rates to know the impact of the two types of LOH. While the I-LOH rates were $3.1 \times 10^{-5}$ per site per division and $5.6 \times 10^{-2}$ events per division, we found that the T-LOH rates were $4.2 \times 10^{-5}$ per site per division and $9.2 \times 10^{-3}$ events per division. These results clearly show that if the number of genome sites impacted by the I-LOH and T-LOH events is the same, the number of events is very different as already pointed out previously.

## Large variation of the LOH spectrum across genetic backgrounds

As mentioned previously, variation can be observed both in the number of events and in the fraction of the genome under LOH by examining all of the LOH events accumulated in the 169 MA lines. These observations prompted us to study the variability of LOH accumulation variation across the nine studied genetic backgrounds. Interestingly, we found that the number of accumulated LOH events varied considerably between the strains (**Figure 2A**). While the H1 lines accumulated the fewest LOH events (n = 10.8 on average), the H4 lines accumulated the largest number of events (n = 413.5 on average), which represents a difference of 38.2 times. We then examined this trend according to the type of LOH events (i.e. I-LOH or T-LOH) accumulated in the different genetic backgrounds. In most cases (except for the H1 and H9 lines), the I-LOH events were significantly in excess compared to T-LOH events in all backgrounds (**Figure 2B**). The variation in the LOH spectrum across strains is mainly related to a difference in the number of I-LOH. While there are 7.2 I-LOH events on average in the H1 lines, the H4 and H5 lines accumulated a large number of 395.9 and 294.8 I-LOH events on average, respectively. Although there is a difference in terms of T-LOH between genetic backgrounds, the variance is relatively small. The number varies from 3.6 events on average for the H1 lines to a maximum of 60.9 T-LOH events on average for the H9 lines. The frequency of I-LOH events was elevated on large chromosomes and a positive correlation with the chromosome size was

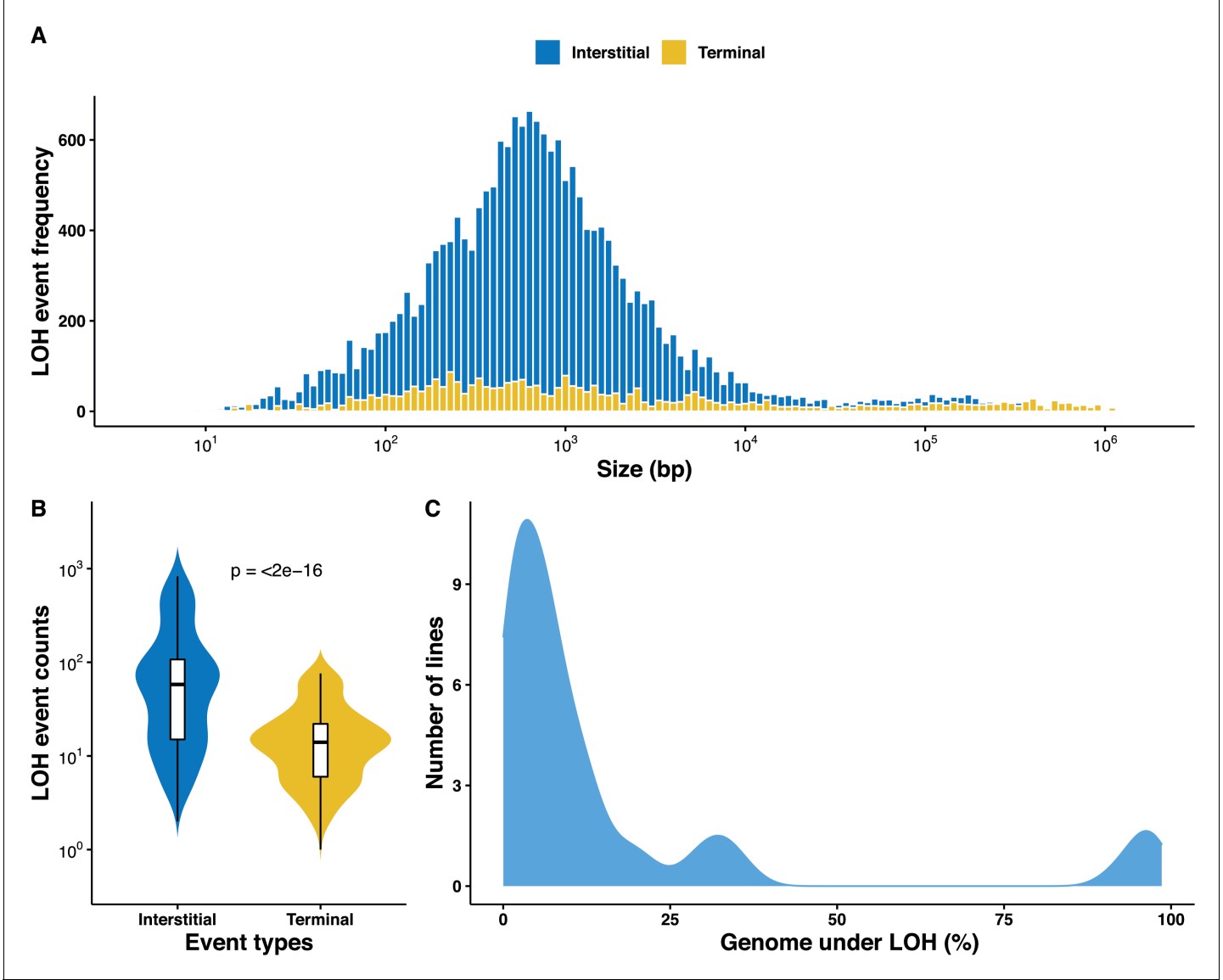

**Figure 1.** Overall distribution of loss of heterozygosity (LOH) in the 169 mutation accumulation (MA) lines. (A) LOH event tract size distribution across all 169 MA lines, the average tract sizes of the interstitial LOH (I-LOH) events (7.4 kb) and terminal LOH (T-LOH) events (55.3 kb), respectively. The global average LOH event size was 14.1 kb. (B) Violin plot of the LOH event counts in the MA lines population, I-LOH events were found to be significantly greater than T-LOH events (Wilcoxon test, $p<2\times10^{-16}$). (C) Distribution of MA lines based on the proportion of genome under LOH (%), dashed line indicates average proportion of genome under LOH across the 169 MA lines, 15.9% (±1.86).

The online version of this article includes the following figure supplement(s) for figure 1:

**Figure supplement 1.** Distribution of the heterozygous single nucleotide polymorphism (SNP) densities as a fraction of total heterozygous SNPs in 5 kb windows across the ancestral diploids as described in *Table 1*.

**Figure supplement 2.** Chromosome-wide distribution of loss of heterozygosity (LOH) events across all mutation accumulation (MA) lines.

**Figure supplement 3.** A terminal LOH (T-LOH) events are significantly larger than interstitial LOH (I-LOH) in the 169 mutation accumulation (MA) lines all backgrounds except H4 (Wilcoxon test, *p < 0.05; **p < 0.01; ***p<0.001; ****p < 0.0001; ns – not significant).

**Figure supplement 4.** Boxplot depicting the fraction genome under loss of heterozygosity (LOH) in the nearly homozygous lines is significantly greater than the rest of the mutation accumulation (MA) lines (Wilcoxon test, $p=6.1\times10^{-10}$).

**Figure supplement 5.** Chromosome-wide loss of heterozygosity (LOH) plots across representative lines from H1, H3, H4, and H9.

observed for all backgrounds except H1 and H9 (*Figure 2—figure supplement 2*). The overall fraction of I-LOH events was significantly increased by more than 10–22 times in the H2, H4, and H5 lines, while in the others, the I-LOH events were only 1.3–2.7 times more frequent than the T-LOH events. These variations in the ratio between the I-LOH and T-LOH events are similar to those

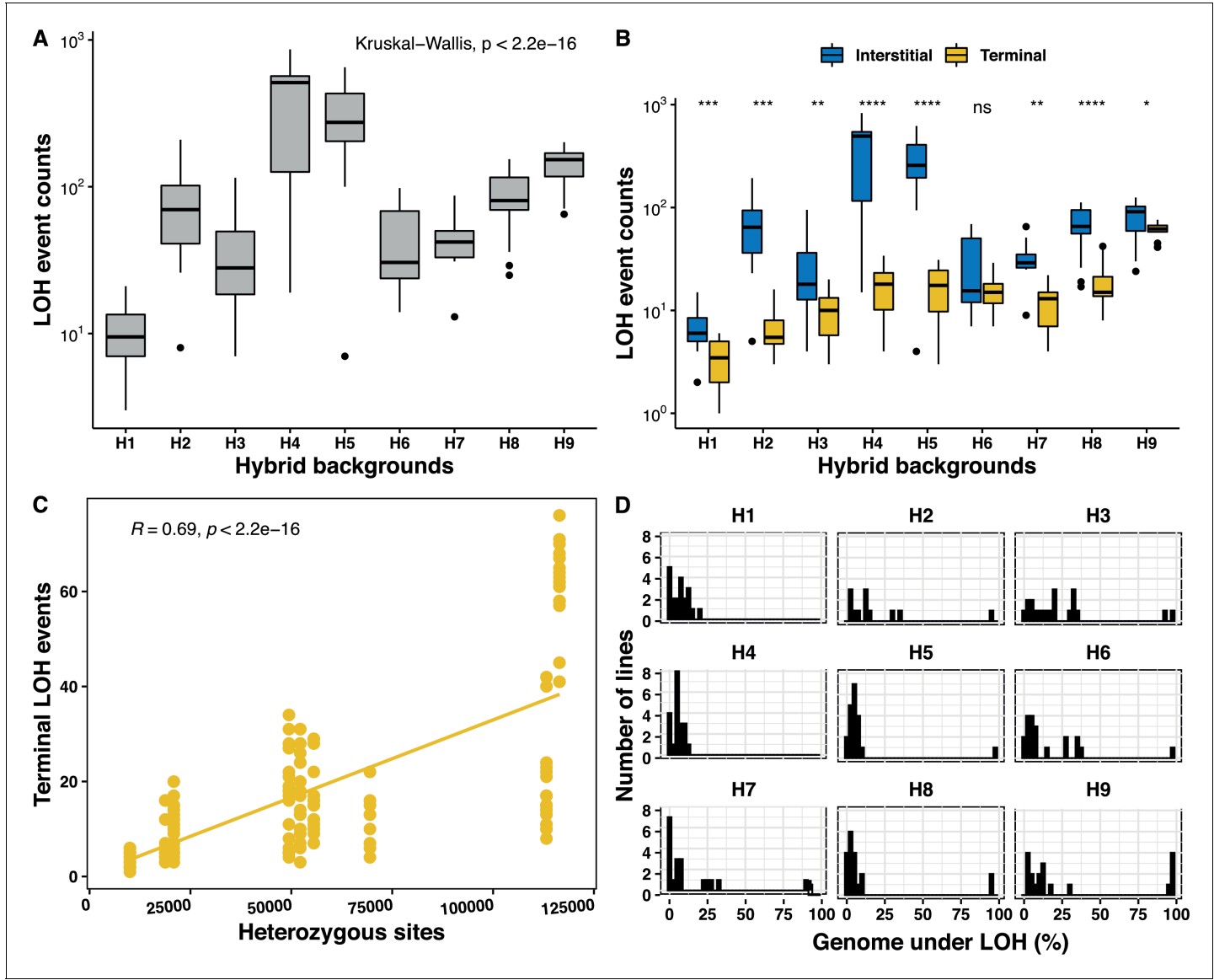

**Figure 2.** Genetic background-dependent variation in the loss of heterozygosity (LOH) repertoire. (A) The frequency of total LOH events across the nine genetic backgrounds are highly variable (Kruskal-Wallis test, $p<2\times10^{-16}$). (B) Variability in interstitial and terminal LOH events counts across all backgrounds H1–H9 (Kruskal-Wallis test, $p<10^{-16}$), interstitial events were always in excess, except for in H6 (Wilcoxon test, *$p < 0.05$; **$p < 0.01$; ***$p<0.001$; ****$p < 0.0001$; ns – not significant). (C) Frequency of terminal LOH events increases with increasing heterozygosity in the nine genetic backgrounds (Pearson's correlation; $r = 0.69$, $p<2\times10^{-16}$), interstitial and total events do not bear any correlation with the background heterozygosity (Pearson's correlation; $p>0.05$). (D) Proportion of genome under LOH is significantly variable across the nine backgrounds (H1–H9) (Kruskal-Wallis test, $p<2\times10^{-16}$).

The online version of this article includes the following figure supplement(s) for figure 2:

**Figure supplement 1.** The frequency of interstitial LOH (I-LOH) events increases with increasing chromosome size for all backgrounds except for H1 and H9; no such correlation was observed for terminal LOH (T-LOH) events.

**Figure supplement 2.** Overall fraction of interstitial and terminal loss of heterozygosity (LOH) tracts across the nine genetic backgrounds.

**Figure supplement 3.** Fraction of the genome fixed toward either of the parental genomes across the nine genetic backgrounds.

observed in various DNA repair and recombination mutants in *S. cerevisiae* hybrids (*Figure 2—figure supplement 2*; *Sui et al., 2020*; *Loeillet et al., 2020*). Interestingly, we observed that the more the ancestors were heterozygous, the more they accumulated T-LOH events (*Figure 2B*). And we have indeed found a positive correlation between the heterozygosity level and the number of T-LOH

events (*Figure 2C*). Such a correlation was found neither for I-LOH nor for the total number of LOH events accumulated in the lines.

Similar to the number of LOH events, the proportion of the genome under LOH also varies across genetic backgrounds. This fraction varies from 3.6% to 16.6% of the genome across the nine backgrounds, when we exclude the nearly homozygous lines reaching more than 90% of the genome being under LOH (*Figure 2D*). The H3 lines accumulated the most genome under LOH (16.6%) while H8 accumulated the least (3.8%), representing a 4.6-fold difference. In addition, the H9 background was enriched in nearly homozygous lines (n = 5), while H1 and H4 had none (*Figure 2D*). We did not observe a significant excess in fixation of either of the parental genomes in the different backgrounds (binomial test, p>0.05), except for the H4 and H5 lines (binomial test, p<0.05). In both, H4 and H5, more than 80% of the genome under LOH is biased toward one of the parents (*Figure 2—figure supplement 3*), although the total genome under LOH in these backgrounds was very low with a level of 5.5% and 4.8%, respectively.

Finally, we also sought to determine the LOH rate variation across the studied genetic backgrounds. The LOH site rates in these hybrids vary between $2.4 \times 10^{-5}$ and $1.1 \times 10^{-4}$ per site per division, and consequently a low variance was observed (*Supplementary file 3*). By contrast, the LOH event rates range from $5.6 \times 10^{-3}$ to $2.8 \times 10^{-1}$ events per division and are hence much more variable. The variability in event rates stems from I-LOH rates more than T-LOH rates. Indeed, while the T-LOH rates vary from $1.5 \times 10^{-3}$ to $2.5 \times 10^{-2}$ events per division, the I-LOH rates vary from $2.9 \times 10^{-3}$ to $2.2 \times 10^{-1}$ events per division.

## Hybrid spore fertility and LOH accumulation

In our study, we selected ancestors with increasing genetic divergence (from 0.1% to 1%) and therefore with varying progeny viability. With an increase in genetic divergence, the role of anti-recombination as well as the increase in the probability of having genetic incompatibilities leads to a decrease in the viability of the offspring (*Greig, 2009*; *Hou et al., 2016*). By determining the spore viability as a measure of spore fertility in the ancestors, we indeed saw that it varies between 38% and 92% (*Supplementary file 4*). While the H1, H2, H3, H6, and H7 ancestors displayed good spore fertility (~85%), the spore viability of the H4, H5, H8, and H9 ancestors was severely compromised (~50%) (*Supplementary file 4*). We took advantage of this variability to assess the impact of it on the accumulation of LOH events. Interestingly, we observed that lines with high spore fertility accumulated much less LOH events than the lines showing low spore fertility (*Figure 3A*). The frequency in these was 7.6-fold higher per bottleneck. In addition, we also found that the size of the LOH events was on average much shorter in low spore fertility lines (*Figure 3B*). We therefore looked at the variation of the accumulated T-LOH and I-LOH events. In lines with high spore fertility, I-LOH events were three times more frequent than those of T-LOH, similar to what had previously been observed in the W303/YJM789 hybrid (*Sui et al., 2020*). However, I-LOH events were even more frequent (seven-fold) than T-LOH in the low spore fertility hybrids with low spore fertility. Overall, frequent short LOH tracts were a signature of the lines derived from hybrids with low spore fertility, that is, the H4, H5, H8, and H9 lines (*Figure 3—figure supplement 1*). MA lines derived from ancestral high spore fertility diploids accumulate large LOH events. The mean LOH event size in the latter is 45.8 kb compared to 7.4 kb in low spore fertility hybrids. The difference in the size of the events may be indicative of the specific underlying mechanisms. Moreover, meiotic spore fertility and recombination fidelity show a positive correlation in laboratory intraspecific *S. cerevisiae* hybrids (*Raffoux et al., 2018*).

In addition to the ancestors, we also determined meiotic spore viability in all 169 MA lines at the end of the experiment. Sporulation was consistent across all lines after 48 hr on sporulation media, with the exception of H9-8 and H9-11 which did not sporulate (*Supplementary file 4*). The spore viability of the lines derived from the highly fertile hybrids did not deviate significantly from their ancestral levels, meaning that they retained their high spore viability. By contrast, individual lines of the four other backgrounds presenting a low spore fertility (lines H4, H5, H8, and H9) showed significant increase compared to the ancestral spore fertility (*Figure 3C*). Rescue of the spore fertility in most MA lines was gradual, with only 42 out of 77 lines showing statistically significant increases (*Supplementary file 4*). MA lines showed up to a 23% increase in spore fertility and the nearly homozygous lines from all hybrid backgrounds were able to fully restore their spore fertility (~90%). Overall, these results suggest that recovery of spore fertility in the MA lines during mitotic

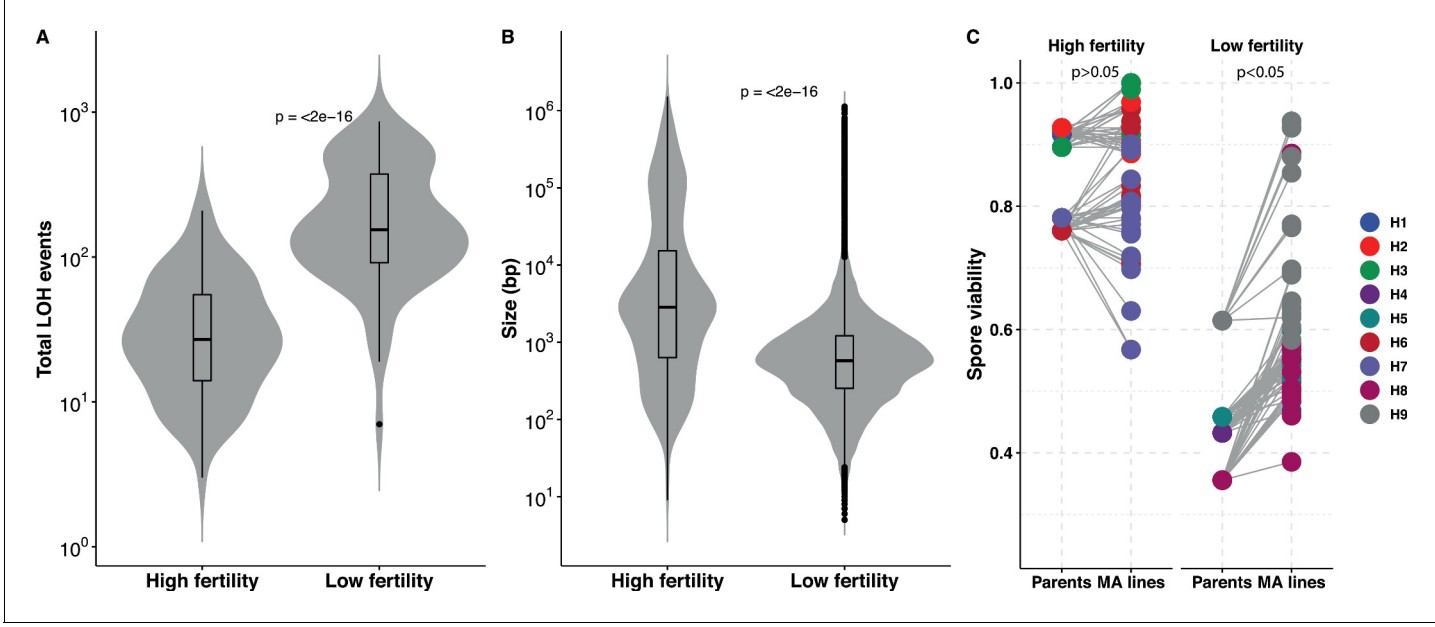

**Figure 3.** Loss of heterozygosity (LOH) accumulation is associated with spore fertility. (**A**) Total LOH events accumulated in mutation accumulation (MA) lines derived from ancestral diploids with high spore fertility, that is, meiotic spore viability greater than or equal to 75% and the fraction of four-spore viable tetrads greater than 50% (H1, H2, H3, H6, H7) is significantly lower than in MA lines derived from ancestral diploids with low spore fertility, that is, meiotic spore viability lesser than 75% and the fraction of four-spore viable tetrads less than 50% (H4, H5, H8, H9) (Wilcoxon test, p<2×10⁻¹⁶). (**B**) The size (in bp) of the LOH events accumulated in MA lines derived from ancestral diploids with high spore fertility is significantly larger than in MA lines derived from ancestral diploids with low spore fertility (Wilcoxon test, p<2×10⁻¹⁶). The average LOH event size in the high spore fertility and low spore fertility MA lines are 45.8 and 7.4 kb, respectively. (**C**) Spore viabilities in both the high and low spore fertility groups compared respective to their ancestral diploids. There is a significant improvement in the spore viabilities of the MA lines derived from the low spore fertility ancestors (Mann-Whitney U test, p=0.04), whereas the viabilities do not change in the MA lines derived from the high spore fertility ancestral diploids (Mann-Whitney U test, p=0.34). The spore viabilities of the individual MA lines and the ancestral diploids detailed in *Supplementary file 4*. The online version of this article includes the following figure supplement(s) for figure 3:

**Figure supplement 1.** The total number of interstitial LOH (I-LOH) and terminal LOH (T-LOH) events are significantly greater in the mutation accumulation (MA) lines derived from ancestral diploids with low fertility (Wilcoxon test, *p < 0.05; **p < 0.01; ***p<0.001; ****p < 0.0001; ns – not significant).

propagation can be achieved either incrementally by undergoing frequent, short LOH events or completely by undergoing rapid whole-genome homozygotization.

## Mutation rate is constant across heterozygous diploid backgrounds

MA experiments have been primarily used to determine mutation rates in a number of model organisms. We therefore estimated mutation rates to understand if the mutational process is also impacted and variable across genetic backgrounds. DNA synthesis during mitotic recombination is considered to be mutagenic (*Pham et al., 2020*) and this effect is suggested to be amplified in heterozygous genomes (*Strathern et al., 1995*; *Rattray et al., 2015*). In the 169 MA lines, we identified 912 single nucleotide mutations (SNMs) and 14 multi-nucleotide mutations (MNMs), of which 83 and 2 are homozygous, respectively (*Supplementary file 5*). MNMs were defined as consecutive SNMs that were not separated by more than one base pair. While 101 SNMs and 2 MNMs were detected in the near-homozygous lines, 811 SNMs and 12 MNMs were identified in the 155 remaining lines. Overall, 70% of the mutations were observed in genic regions and 30% in the non-genic areas, and we observed a significant excess of non-synonymous mutations over synonymous mutations in the genic regions (binomial test, p>0.05). These biases have already been observed in previous yeast MA experiments. Transition mutations were more frequent than transversions with an average Ts/Tv ratio of 1.35. SNM and MNM rates were on average $1.1 \times 10^{-10}$/site/division and $1.98 \times 10^{-12}$/site/division, which is similar to previous estimates in isogenic and hybrid diploid *S. cerevisiae* strains

(*Dutta et al., 2017*; *Sharp et al., 2018*; *Liu and Zhang, 2019*). SNM rates are five orders of magnitude lower than LOH rates.

The proportion of SNMs in the six categories of transitions and transversions varies across the nine backgrounds ($p<10^{-4}$, Chi-square test; *Figure 4—figure supplement 1A*, *Supplementary file 6*). Nevertheless, the transitions are always in excess compared to transversions and we found that the Ts/Tv ratio is always greater than 0.5. The Ts/Tv ratio was similar across the backgrounds with only a significant increase of up to 1.86 for the H3 lines ($p<10^{-5}$; Chi-square test, *Figure 4—figure supplement 1B*, *Supplementary file 6*). Previous studies have reported a mutational bias of GC > AT over AT > GC mutations in yeast. We indeed found a GC > AT/AT > GC bias in all genetic backgrounds, with the exception of the H8 lines and significantly increased compared to the rest in H1 (*Figure 4—figure supplement 1C*, *Supplementary file 6*). The frequency of mutations was more impacted by the neighboring nucleotide on C/G sites than on A/T sites ($p<0.01$, binomial test; *Figure 4—figure supplement 2*, *Supplementary file 6*). Similar variations in the SNM spectrum have also been observed across different yeast species (*Nguyen et al., 2020*).

Finally, we estimated mutation rates in the nine genetic backgrounds to determine if they were as variable as the LOH rates. The SNM rates vary from $0.65 \times 10^{-10}$ to $1.56 \times 10^{-10}$/site/division across the nine heterozygous diploids, representing a 2.4-fold difference. SNM rates were not significantly different within or across the nine backgrounds (*Figure 4*). The average mutation rate in the 14 near-homozygous lines was just slightly higher, around $1.92 \times 10^{-10}$/site/division. Furthermore, we observed no correlation between the mutation rates and the heterozygosity levels, GC content, or LOH rates. Overall, while the spectrum of SNMs is variable, their overall frequency of occurrence remains relatively constant across genetic backgrounds.

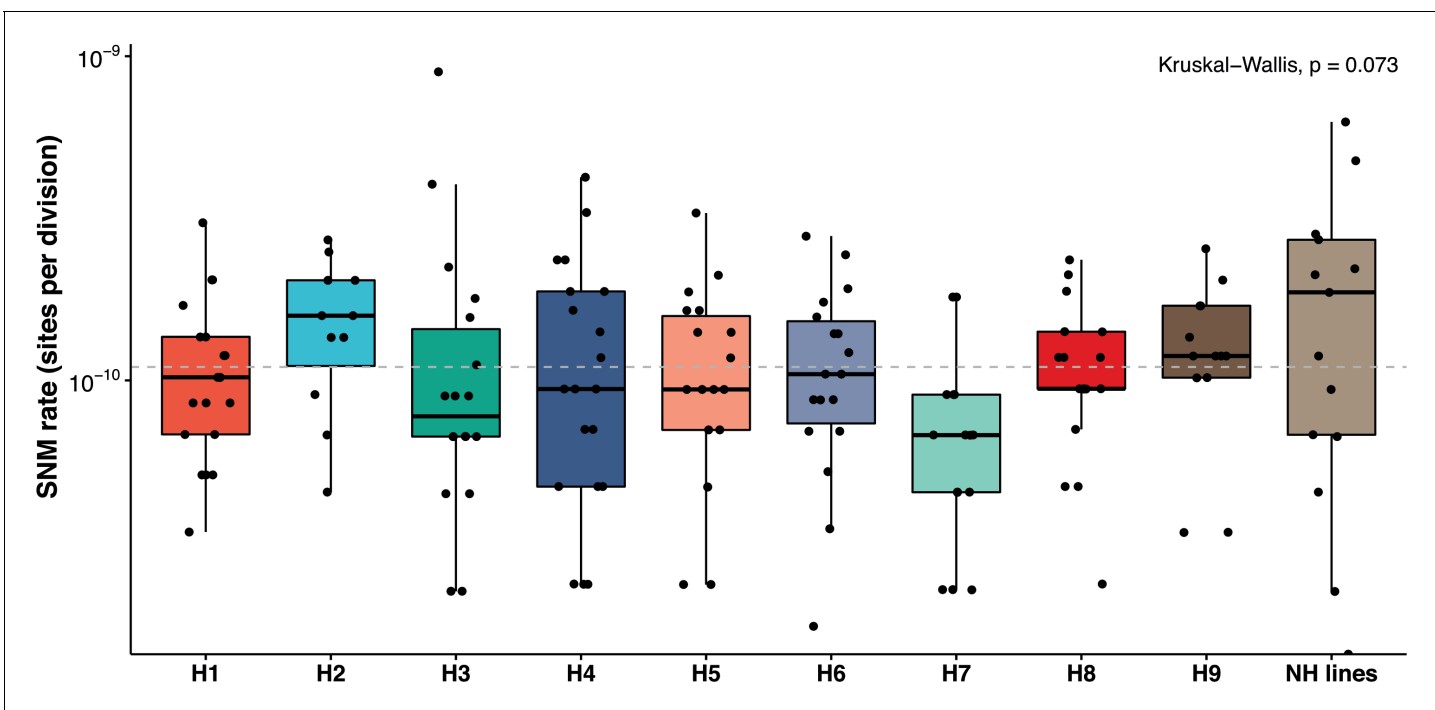

**Figure 4.** Mutation rates are constant across the backgrounds and within the nearly homozygous (NH) lines (range 0.67–1.9 × 10⁻¹⁰ mutations/site/division; Kruskal-Wallis test, p>0.05). The overall, mean single nucleotide mutation (SNM) rate in the 169 MA lines 1.1 × 10⁻¹⁰ per site per division is not different from previous estimates in various diploid *Saccharomyces cerevisiae* strains. NH lines represent the SNM rates in the NH lines. SNMs have been detailed in *Supplementary files 5* and *6*.

The online version of this article includes the following figure supplement(s) for figure 4:

**Figure supplement 1.** SNM spectrum across the nine hybrid backgrounds.

**Figure supplement 2.** The frequency of mutations was significantly impacted at C/G sites than A/T sites by the neighboring nucleotides (Chi-square test, p<0.05; *Supplementary file 5–6*).

**Figure supplement 3.** The frequency of aneuploid chromosomes across the nine genetic backgrounds (H1–H9).

In addition, we also detected 51 aneuploidy events in the MA lines, 43 chromosomal gains and 8 losses (*Figure 4—figure supplement 3*). No aneuploidy events were detected in the H2, H5, and H7 MA lines. Ancestral trisomies have been observed in H3 (chromosome IX) and H9 (chromosomes IX and XVI), and chromosome losses always correspond to the loss of extra copies of chromosomes IX and XVI in these two genetic backgrounds. The overall rate of aneuploidy is $1 \times 10^{-4}$ per division, which is similar to that determined in previous studies (*Sui et al., 2020*; *Loeillet et al., 2020*; *Sharp et al., 2018*; *Zhu et al., 2014*). The LOH and SNM rates are not different among the euploid and aneuploid MA lines or the nearly homozygous lines (Mann-Whitney U test, p>0.05).

## Mitotic propagation can lead to rapid homozygotization

As mentioned previously, we found a total of 14 nearly homozygous lines exhibiting more than ~90% of the genome under LOH, although it varies from 88% to 98% (*Figure 1—figure supplement 5*). These lines were present in all the genetic backgrounds except for two (H1 and H4) and with a prevalence in the H9 background. The LOH spectrum in these lines is characterized by a low number of events overall with similar frequency of I-LOH and T-LOH events. The average LOH event size is 185.1 kb, significantly larger than the rest of the 155 lines (~7 kb). The nearly homozygous lines coming from low spore fertility backgrounds were able to fully restore their spore fertility (~90%) (*Supplementary file 4*).

We also determined the mitotic growth rates of the MA lines at the beginning and the end of the MA experiments (*Supplementary file 2*). Interestingly, a decrease in the growth rates on average was observed at the end for the lines, but this decrease was significantly greater for the near-homozygous lines compared to the other lines (Mann-Whitney U test, $p<10^{-10}$). Gain in spore fertility appears to be associated with decline in vegetative growth. Life history trade-offs involving asexual and sexual reproduction under nutrient stress were found to be correlated in *S. cerevisiae* (*Magwene et al., 2011*). Furthermore, the rates of SNM and MNM in these 14 lines were 1.8-fold and 1.5-fold greater, respectively. Relatively higher mutation rates may be responsible for the higher decrease in growth rates. The frequency of homozygous mutations was found to be 3.2 times higher compared to the rest of the MA lines, but most of the detected mutations in these lines were heterozygous (*Supplementary file 5*). Since most new SNMs are heterozygous, most SNMs in the nearly homozygous lines may have been accumulated after the homozygotization event and mutation rates might have been higher (*Nishant et al., 2010*).

## Discussion

Mitotic recombination events are thought to occur with ~$10^5$-fold less frequency than meiotic recombination and therefore require a selection system for their detection (*Lee et al., 2009*). Over a large number of divisions, such changes may accumulate to create a significant impact on the genome that can be detected and is relevant to measure, given the ratio between mitotic and meiotic cycles in yeast (*Ruderfer et al., 2006*; *Kelly et al., 2012*). Given the role of LOH events in evolution and disease, it is important to understand the extent of LOHs over a large number of mitotic divisions. We therefore propagated MA lines across different heterozygous diploid hybrids over many generations to probe the impact of genetic background on the LOH landscape in *S. cerevisiae*. Our study sheds light on cumulative genotypic changes across the MA lines that were propagated vegetatively. We surveyed a large set of 22,828 LOH events to accurately determine variations in LOH patterns and their subsequent implications with SNM rates and spore fertility. Overall, we observed an extensive accumulation of LOH events but a spectrum that was highly variable depending on the studied genetic backgrounds.

Our MA lines show significant variation both in the types of LOH events accumulated and in the rates. Studies involving homozygous deletion screen of non-essential genes in the BY diploid strain have identified several gene deletion backgrounds involving DNA repair, chromatin assembly, and kinetochore function with locus-specific effects on LOH associated with recombination and whole chromosome loss (*Yuen et al., 2007*; *Andersen et al., 2008*). A previous MA study also highlighted a difference in LOH accumulation in two different hybrids (S288c/YJM789 and RM11/S288c) showing a similar heterozygous level (*Dutta et al., 2017*; *Pankajam et al., 2020*). Conversely, another study-involving serial passages of two hybrids with different levels of heterozygosity showed no difference in the accumulation of LOH (*James et al., 2019*). Here, we decided to investigate many

heterozygous hybrids with a wide variation in the levels of heterozygosity. The fraction of the genome under LOH varies considerably across the nine diploid hybrids studied. Gene conversions contribute toward 86% of the events (I-LOH) while mitotic CO and BIR make up the rest (T-LOH) (*Ene et al., 2018*). Whereas I-LOH events are more evenly distributed across the chromosomes, T-LOH events are enriched at the ends of chromosomes (*Sui et al., 2020*). LOH rate variation has also been observed in *Cryptococcus* sp. and *Candida albicans* in response to changing environments, such as high temperatures and the presence of antifungal drugs (*Bouchonville et al., 2009*; *Li et al., 2012*; *Forche et al., 2018*). In our MA lines, I-LOH events are over six times more frequent than the T-LOH events and differentially distributed along the chromosomes (*Figure 1—figure supplement 2A–C*). These differences may result from the initiation of DNA damage and repair mechanisms involved. Recombination and conversion hotspots are not conserved during meiotic DSB repair and the same may be true for mitotic DSB repair (*Sui et al., 2020*; *Mancera et al., 2008*). In addition, induction of mitotic DSBs is not synchronized and therefore, the recruitment of the repair machinery may also be spatially and temporally differentiated (*Whelan et al., 2018*; *Vítor et al., 2020*).

The frequency of short I-LOH events was higher in backgrounds with low ancestral spore fertility, that is, the H4, H5, H8, and H9 lines. Conversion events triggered by mismatch repair (MMR) can be short, often below 1 kb, as it can be the case in low spore fertility hybrids (*Palmer et al., 2003*). The MMR machinery stifles recombination between divergent sequences, preventing events such as chromosomal rearrangements, deleterious mutations, and thus leading to reproductive isolation (*Petit et al., 1991*; *Myung et al., 2001*). Genetic incompatibilities with the potential to pose reproductive barriers are widespread within species (*Corbett-Detig et al., 2013*). Disruptions in the MMR pathway can rescue hybrid sterility and have subsequently been shown to enhance LOH accumulation in both intra- and interspecific yeast hybrids during RTG (*Greig, 2009*; *Hunter et al., 1996*; *Greig et al., 2002*; *Mozzachiodi, 2020*). Regarding the T-LOH events, we observed a heterozygosity-dependent increase in the frequency of these events across the nine backgrounds. Heterozygosity can enhance recombination-mediated repair of DSB (*Clikeman et al., 2001*). Moreover, the levels of heterozygosity as well as variability of the distribution of polymorphisms along the genome significantly impact CO frequencies during *Arabidopsis thaliana* meiosis (*Ziolkowski et al., 2015*).

Hybridization is a process common to the life history of many organisms and rapidly generates genomic diversity (*Eberlein et al., 2021*; *Faulks and Östman, 2016*; *Vallejo-Marin and Lye, 2013*). Even though new hybrids tend to exhibit a low spore fertility, natural hybridizations have played an essential role in the diversification and evolution of the genome of many fungal, plant, and animal species. Hybrid sterility arises from events such as genomic incompatibilities, chromosomal rearrangements, and ploidy changes. Restoring spore fertility is an essential step toward establishing a new population or species. In our MA line experiment, ancestral diploids show offspring viability ranging from 35% to over 90%. We have observed that LOH accumulation is associated with a restoration of spore fertility. While the MA lines derived from low spore fertility hybrids show a gradual return of spore fertility, the nearly homozygous lines were able to fully restore their spore fertility. Influx of homozygosity due to LOH may result in less heteroduplex rejection during recombination repair of DSBs (*D'Angiolo et al., 2020*). Improved repair and fewer genetic incompatibilities can increase spore viability in homozygous backgrounds. Polyploidization events during vegetative propagation in highly diverged *Saccharomyces paradoxus* hybrids have also been associated with spore fertility recovery (*Charron et al., 2019*). However, the near-homozygous lines do not exhibit significant deviations from diploidy. The near-homozygous lines appear to undergo an accumulation of LOH events via mitotic recombination-independent mechanisms. The size and distribution of the LOH events in these lines are not consistent with the results of mitotic recombination. Mechanisms such as chromosome loss and re-duplication or hyper-recombination could be at the origin of the appearance of these near-homozygous lines. In addition, it has been shown that *S. cerevisiae* replication and DNA repair mutants (such as *rad27*) can lead to rapid homozygosity covering up to 60% of the genome after 25 vegetative bottlenecks (*Loeillet et al., 2020*). Alternatively, these may be due to multiple rounds of RTG, that is, mitotic repair of Spo11-induced DSBs during aborted meiosis (*Dayani et al., 2011*; *Laureau et al., 2016*; *Mozzachiodi, 2020*). Nevertheless, none of the ancestral diploids or end MA lines sporulate on rich media.

MA experiments have been a popular method for measuring mutation rates in model organisms. Mutation rate estimates are available for haploid and diploid yeast genomes during meiotic and

vegetative propagation (*Lynch et al., 2008*; *Sui et al., 2020*; *Dutta et al., 2017*; *Zhu et al., 2014*; *Nishant et al., 2010*). Ploidy has a significant impact on the mutational process in yeast and the diploid genome is buffered against chromosomal instabilities (*Sharp et al., 2018*; *Nishant et al., 2010*). LOH events have been underappreciated compared to single-base mutations and large alterations such as deletions, duplications, translocations, and changes of the ploidy level. This is mainly due to the fact that most MA line experiments were performed on isogenic or homozygous model isolates. Recent interest in heterozygous genomes has indicated that LOH may have a broader and more significant impact on genome evolution. Genomic heterozygosity can also alter the mutational process (*Strathern et al., 1995*; *Rattray et al., 2015*). Different environments have also been shown to impact mutation rates in *S. cerevisiae* (*Liu and Zhang, 2019*) and mutation rates were found to be different across seven distinct yeast species (*Nguyen et al., 2020*). Here, the genetic background impact was limited to the mutation spectrum. We did not observe any significant difference in the overall SNM rates in the derived MA lines. Neither the genetic background nor the inherent heterozygosity appears to have a significant impact to the overall mutational process. The only variability in mutation rates observed among genetic backgrounds comes from mutator alleles, which may be rare in the *S. cerevisiae* species. Interestingly, the SNM rates in the nearly homozygous lines were higher but not significantly different from the rest, and the SNMs were mostly heterozygous. These observations suggest that most of the mutations in these lines appeared after the homozygotization events and that the rates may have been underestimated, compared to the other lines.

## Materials and methods

### Strain construction and MA lines propagation

The *S. cerevisiae* strains used in this study are described in *Supplementary file 1*. Yeast strains were grown on either YPD (yeast extract 1%, peptone 2%, dextrose 2%) or synthetic complete SC (yeast nitrogen base 0.67%, amino acid mix 0.2%, dextrose 2%) medium at 30℃ (*Mortimer and Johnston, 1986*; *Rose, 1990*; *McCusker et al., 1994*). Briefly, haploid strains of opposite mating (*MATa, ho:: KanMX; MATα, ho::NatMX*) types were crossed on SC for 6 hr. These were streaked down to single colonies on YPD medium supplemented with G418 (Euromedex – 4ml/l of 50 mg/ml stock) and nourseothricin (Jena Bioscience – 2 ml/l of 50 mg/ml stock) to select for diploids (*Goldstein and McCusker, 1999*).

Eight diploid colonies were isolated from each of the crosses H1 through H9 and ploidies were confirmed by flow cytometry and two diploid colonies were frozen. These were also checked for their ability to sporulate on 1% potassium acetate agar. One frozen stock of each of the hybrid backgrounds was designated as the ancestor and further streaked down to single colonies. In total, 20 single colonies from each of the diploid hybrid were isolated and propagated. Individual replicate lines were bottlenecked to a single colony every 48 hr for at least 75 bottlenecks on YPD agar. Intermediates were frozen down every 25 bottlenecks until the end of the experiment. MA lines at the end of the experiment were sequenced.

### Growth rates and tetrad dissection

All MA lines and ancestral diploids were revived on YPD agar plates and further streaked down to single colonies. After 24 hr, three independent average sized colonies per line were picked in 1 μl of ddH$_2$O. The total number of cells (N) counts were estimated using a cell counter and subsequently, growth rate per hour (r) was calculated assuming exponential growth described by N = e$^{rt}$. The generation time was estimated as g = ln2/r per hour and the number of divisions per day was 24/g = 24 r/ln2 (*Liu and Zhang, 2019*). Diploid ancestors and MA lines after 100 bottlenecks were revived on YPD agar and patched on sporulation media (1% potassium acetate agar) for up to 72 hr (*Argueso et al., 2004*). Tetrad dissections were performed using the SporePlay microscope (Singer Instrument) on DIFCO YPD agar and at least 24 and 48 tetrads were dissected for the high fertility and low fertility groups, respectively.

### Whole-genome sequencing of diploid strains

Genomic DNA was extracted from the 169 MA lines using the Omega yeast DNA kit (Life Science Products) and DNA libraries were prepared from 5 ng of total genomic DNA using the NEBNext

Ultra II FS DNA Library kit for Illumina (New England Biolabs) following manufacturer's protocols. Following quality check using a Bioanalyzer 2100 (Agilent Technologies) and quantification using the Qubit dsDNA HS assay, 4 nM of each of the libraries were pooled and run on a NextSeq 500 sequencer with paired-end 150 bp reads by the EMBL Genomics Core Facility (Heidelberg, Germany).

## Read mapping, genotyping of sequencing data

Sequencing reads from Fastq files were mapped to the masked (RepeatMasker, default parameters, masking simple repeats, and low complexity regions) *S. cerevisiae* R64 reference genome using bwa mem (v0.7.17). Resulting bam files were sorted and indexed using SAMtools (v1.9). Duplicated reads were marked, and sample names were assigned using Picard (v2.18.14). GATK (v3.7.0) was used to realign remaining reads. Candidate variants were then called using GATK UnifiedGenotyper. The calling was done simultaneously for lines from the same background.

## Analysis of LOH tracts

After variant calling, SNPs called in each hybrid parental couple were first filtered (bcftools v1.9) to define a set of confident markers expected to be heterozygous in the hybrid progeny. Positions with a single alternate allele, supported by at least 50 sequencing reads across both parents, were kept as parental markers. Joint SNP calling was performed background-wise as described previously. Bcftools isec was used to extract SNPs located at parental markers positions in all samples. In case of homozygosity at markers position, a parental origin tag was added by comparing the allele to parental alleles. Positions with GQ less than 20 were filtered out. LOH events were called using an in-house script. Heterozygous and LOH tracts from either parental origin were initially defined by uninterrupted tracts of successive marker positions with the right tag (heterozygous, parent1, parent2). Single-marker tracts were filtered out and subsequent tracts from the same origin were merged. Any tracts with 80% or more overlap and shared by at least 50% of the lines from the same hybrid were excluded. Average LOH tract coordinates were determined as the mean between the coordinates of first or last marker of a given tract and the first previous or next marker around that tract. LOH tracts were tagged as terminal if they overlapped the first or last 20 kb of a chromosome and tagged as interstitial otherwise. LOH site rates per line were calculated as $N/D*G$, where $N$ = sites under LOH (I-LOH, T-LOH or overall), $D$ = total number of divisions, and $G$ = total genome size. LOH event rates per line were calculated as $N/D$, where $N$ = number of events (I-LOH, T-LOH, or total events) and $D$ = total number of divisions. The LOH plots as in *Figure 1—figure supplement 5* were generated using the tool *karyoploteR* (*Gel and Serra, 2017*).

## Analysis of new mutations

Filters were applied to the SNPs that were called from the MA lines sequencing results in order to identify SNPs that were exclusive to a line, that is, mutations that occurred during the MA experiment (*Sharp et al., 2018*; *Liu and Zhang, 2019*; *Zhu et al., 2014*). An initial filtering round was first applied, and only positions covered by more than 10 reads in each sample with a single alternate allele were kept. Then, filters based on the numbers of lines per background and the type of conversion event expected to occur in a given background (homozygous to homozygous, homozygous to heterozygous) were applied to positions that were initially in a homozygous state (homozygous to homozygous, homozygous to heterozygous). Bcftools (v1.9) was used to perform the filtering. For homozygous-to-homozygous conversions, SNPs occurring within a called LOH were filtered out as they were actually representing the LOH and not an SNM. The remaining set of SNMs was then analyzed using snpEff to classify intergenic and genic mutations, and for the latter, synonymous and non-synonymous ones. Mutation rates per line were calculated as $N/D*2G$, where $N$ = number of mutations (SNM or MNM), $D$ = total number of divisions, and $G$ = genome size (genome size was multiplied by 2, to account for the diploid genome). Homozygous mutations were multiplied by 2 in the estimation of the rates.

## Acknowledgements

This work was supported by the European Research Council (ERC Consolidator Grant 772505). JS is a Fellow of the University of Strasbourg Institute for Advanced Study (USIAS) and a member of the Institut Universitaire de France.

## Additional information

### Funding

| Funder | Grant reference number | Author |
|---|---|---|
| European Research Council | 772505 | Joseph Schacherer |

The funders had no role in study design, data collection and interpretation, or the decision to submit the work for publication.

### Author contributions

Abhishek Dutta, Conceptualization, Formal analysis, Investigation, Writing - original draft, Writing - review and editing; Fabien Dutreux, Formal analysis; Joseph Schacherer, Conceptualization, Formal analysis, Supervision, Funding acquisition, Investigation, Writing - original draft, Writing - review and editing

### Author ORCIDs

Abhishek Dutta https://orcid.org/0000-0003-2256-6956
Fabien Dutreux https://orcid.org/0000-0003-2569-1330
Joseph Schacherer https://orcid.org/0000-0002-6606-6884

### Decision letter and Author response

Decision letter https://doi.org/10.7554/eLife.70339.sa1
Author response https://doi.org/10.7554/eLife.70339.sa2

## Additional files

### Supplementary files

• Supplementary file 1. Haploid strains used to generate the nine hybrid diploids, standardized names as per *Peter et al., 2018*.

• Supplementary file 2. Mean growth rate estimates at bottleneck 0 and at the end of the experiment for the nine genetic backgrounds. These estimates were used to determine the number of divisions at every bottleneck (see Materials and methods).

• Supplementary file 3. All loss of heterozygosity (LOH) tracts across the 169 mutation accumulation (MA) lines supported by at least two single nucleotide polymorphisms (SNPs). Tracts were merged if consecutive tracts were disrupted by less than two SNPs. LOH tracts in nearly homozygous lines highlighted in yellow.

• Supplementary file 4. Meiotic spore viability in the mutation accumulation (MA) lines, H9-8 and H9-11 excluded as they did not sporulate. SV-percentage spore viability in the MA lines; statistical significance of differences in spore viability between ancestral diploid and the derived MA lines were determined using the p-values from Fisher's exact test (GraphPad prism). N – number of tetrads analyzed for spore viability, ns – not significant; p>0.05.

• Supplementary file 5. Mutations detected in the 169 mutation accumulation (MA) lines. Multinucleotide mutations (MNMs) highlighted in purple.

• Supplementary file 6. Chi-square test p-values for the single nucleotide mutation (SNM) spectrum variation. Chi-square test was performed using the Microsoft excel function Chisq.test().

• Supplementary file 7. List of strains used in this study.

• Transparent reporting form

## Data availability

All strains listed in Supplementary file 7 are available upon request. Sequence data are available from National Centre for Biotechnology Information Sequence Read Archive under accession number: PRJEB43186.

The following dataset was generated:

| Author(s) | Year | Dataset title | Dataset URL | Database and Identifier |
|---|---|---|---|---|
| Dutta A, Dutreux F, Schacherer J | 2021 | Sequencing data from 169 *S. cerevisiae* mutation accumulation lines | https://www.ncbi.nlm.nih.gov/sra/?term=PRJEB43186 | NCBI Sequence Read Archive, PRJEB43186 |

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
