## [Decision Letter]

[Editors' note: this paper was reviewed by Review Commons.]

**Acceptance summary:**

Loss-of-heterozygosity is a phenomenon whereby a piece of sequence of one homologous chromosome is replaced by the corresponding sequence of the other homolog. Using strains of the budding yeast *Saccharomyces cerevisiae* with various genetic backgrounds, the manuscript by Dutta et al. describes the types and frequency of loss-of-heterozygosity events in multiple mutation accumulation lines under standard growth conditions (no stress). Although loss-of-heterozygosity events have been well documented in yeasts, especially in Candida pathogenic yeasts in response to drugs, this study is novel in that it approaches the study of loss-of-heterozygosity from an evolutionary perspective, showing that such events are surprisingly common during asexual reproduction and highly variable across strains.

---

## [Author Response]

Reviewer #1Loss-of-heterozygosity is a phenomenon whereby different mechanisms of DNA repair can drive a heterozygous region to fixation by replacing a piece of sequence of one homologous chromosome by the corresponding sequence of the other homolog. This phenomenon is known to be a significant player in genome diversification and evolution, but quantitative assessment of its role for the dynamics of biodiversity and evolution has little been experimentally addressed so far. Neither had the effect of the genetic background on LOH been widely investigated. This paper reports on a mutation accumulation experiment designed to investigate variations in LOH patterns across different *S. cerevisiae* hybrids, and their relationship with mutation and meiotic spore fertility. Starting from 9 hybrids from distantly-related strains chosen to maximize the ranges of genetic diversity and heterozygosity explored, the authors produced a total of 169 MA lines after at least 1,769 generation of mitotic growth including 75 or more single-cell bottlenecks followed by about 24 divisions each. Final lines were then sequenced to detect, quantify, and characterize LOH events genome-wide, and also to detect, quantify, and characterize new single-nucleotide (SNM) or multi-nucleotides (MNM) mutations.22,828 LOH events were found, distributed in two categories with similar levels of impact (in terms of number of sites under LOH): a majority of short-tract interstitial events (I-LOH), mostly far from chromosome ends, likely due to gene conversion events, and a minority of long-tract terminal events (T-LOH), spanning till the end of the chromosome arm, likely due to mitotic crossovers or BIR. LOH extent across lines varied from almost none to almost the whole genome length, with a strong effect of the genetic background and of the heterozygosity level of the initial hybrid. LOH frequency and extent were significantly different across hybrids, mainly due to different numbers of I-LOH. T-LOH events were associated with higher hybrid heterozygosity, but not I-LOH. An outlier group of 14 lines derived from 7 different hybrids showed an enormous level of LOH (more than 90 %) significantly higher than the rest. Their meiotic spore viability was fully restored during the MA generations and this restoration was associated with a decrease in vegetative growth rate, showing an apparent trade-off. Overall, lines derived from hybrids with initial low meiotic sporulation rate improved their sporulation ability along mitotic generations, either by frequent short LOH events or by rapid whole-genome fixation.Analysis of mutations showed 912 SNMs and 12 MNMs, with a significant excess of non-synonymous mutations in genic regions, showing mutation spectra rather similar to those observed in previous studies in yeast. Mutation rates were not significantly different across lines derived from the nine hybrids. Overall, LOH contributed to a much larger extent to genome diversification than mutation.The results presented in this manuscript are clear and the conclusions are convincing and well supported by facts. The experiments are carefully designed and well described so they can be reproduced. In particular, investigations of LOH or mutation spectra are thoroughly carried out and provide very detailed information. Finally, the results are clearly discussed and compared to previous studies related to the topic. Overall, the paper is well-written and the figures and tables clear and informative.1. I have only one major comment. Because this work tries to relate the types and extent of LOH during mitosis with meiotic spore fertility, I may have expected some part in the introduction and/or discussion about the comparison between mitotic and meiotic processes involved in double-strand break formation, break-induced replication, crossovers, and gene conversion mechanisms, particularly with regards to their possible control by the genetic background and/or by the extent of heterozygosity. This may shed more light on the relationships between LOH and spore fertility, and thus indicate to what extent LOH can be seen as a quick and efficient mechanism to restore sexual fertility after hybridization events between distantly-related parents. Possible involvement of return-to-growth events, their consequences and the expected frequency of such meiosis abortion in nature could also be presented from a general point of view.

We have incorporated text in the introduction (lines 89-98 and lines 105-111) on the role of DSB repair during mitosis and meiosis as well as meiotic abortions in the generation of LOHs. The role of abortive meiosis has also been discussed at lines 422-424.

2. The abstract could mention the number of hybrids (9).

We have now added the number of studied genetic backgrounds in the abstract.

3. P8 'Moreover, meiotic fertility and recombination fidelity show a positive correlation in laboratory intraspecific *S. cerevisiae* hybrids': I'm not sure 'recombination fidelity' is the right term here. Would 'recombination rate' be better ?

We feel that the term ‘recombination fidelity’ is specifically used to reflect both the rate and efficiency of recombination. Recombination rate would just reflect the number of recombination events. We therefore decided to keep ‘recombination fidelity’.

4. P5 The sentence: 'Selection is known to be minimal as the genomic changes accumulated are purely a consequence of drift' may be reformulated with something like: 'Selection is supposed to be minimal as the experiment was designed to make drift by far the main cause of genomic changes'.

We changed the sentence accordingly (lines 150-151).

5. P9 'Overall, 70% of the mutations were observed in genic regions and 30% in the non-genic areas, and we observed a significant excess of non-synonymous mutations over synonymous mutations in the genic regions, indicating no bias in mutation accumulation': To make this point clearer, it would be interesting to detail explicitly which possible biases are considered here.

These biases have been established in a plethora of MA experiments in yeast (see Zhu et al. 2014; Sharp et al. 2018; Liu and Zhang 2019). In fact, we are just observing the same biases and trends in the mutations detected in our MA experiment, therefore our experimental design does not impact the mutation accumulation process. We have reworded the sentence to clarify this point (lines 293-294).

6. P9 'We found that the Ts/Tv ratio is always greater than 0.5': Did the authors mean >1? Please clarify.

The Ts/Tv ratio should be 0.5 under no mutational bias. The transition frequencies are always greater than the transversion frequencies and therefore the Ts/Tv ratios are always >0.5 and often >1 (see, Liu and Zhang 2019).

7. P10 'The nearly homozygous lines coming from low fertility backgrounds were able to fully restore their fertility fully (~90%)': 'fully' is present twice in the sentence. Also, 'fertility' should be replaced by 'spore fertility' or 'meiotic fertility' or something equivalent in that sentence and throughout the manuscript.

The repetition of “fully” has been removed (line 338). We used the term “fertility” because in yeast it has been used extensively to describe meiotic spore viability (See, Greg 2009). However, we agree and to be more precise, we have replaced “fertility” with “spore fertility” throughout the text.

8. P15 'Positions with GQ<20 was filtered out': were filtered out, P15 'Any tracts with 80% or more overlap shared by at least 50% of the clones from the same line were considered as events were excluded': Please correct the sentence.

Both the corrections have been incorporated in the text (lines 500, 499-501).

9. P15 'LOH tracts were tagged as external if they overlapped the first or last 20kb of a chromosome and tagged as internal otherwise (File S1)': Did the authors mean 'interstitial' by 'internal'? Same for 'external' / 'terminal'? Please clarify. I did not understand why it is referred here to File S1 – which is python source code.

We apologize for this error, the terms ‘internal and ‘external’ have now been replaced by ‘interstitial’ and ‘terminal’, respectively (lines 503-505). The reference to the file S1 has also been removed.

Reviewer #2In this manuscript, Dutta et al. analyzed 169 heterozygous *S. cerevisiae* mutation accumulation (MA) lines and investigated the patterns of loss of heterozygosity in them. They measured I-LOH and T-LOH events among them and presented a correlation between LOH and fidelity.1. These analyses are potentially interesting, but the manuscript falls short in presenting the data in the context of current LOH literature. In particular, there have been several MA studies on LOH (in yeast and other model species), unfortunately, there is no clear comparison on the LOH analysis. For instance, how this study will help advance the understanding of LOH, confirm and/or reconcile previous studies. Finally, the presentation of the manuscript needs some serious work. It is hard to follow. The authors should make it concise, yet with needed background information. I cannot recommend it for publication before all these are addressed.

Where possible, we have presented our results in the context of previous literature (see Lynch et al. 2008; Zhu et al. 2014; Sharp et al. 2018; Dutta et al. 2017; James et al. 2019; Ene et al. 2018; Sui et al. 2020; Loeillet et al. 2020). Nevertheless, it is not really possible to ‘confirm and/or reconcile’ our results with previous studies because the aspect of LOH we are looking at has never been explored on this scale. Our results lead to a new insight into the variable dynamics of LOH accumulation across distinct genetic backgrounds. We have further incorporated several changes to improve the text overall (please also see response to reviewer 1 comments).

2. In the abstract. they wrote "We observed that the interstitial LOH (I-LOH) events, resulting in internal short LOH tracts, were much frequent (n = 19,660) than the terminal LOH (T-LOH) events, i.e., tracts extending to the end of the chromosome (n = 3,168)". A statement like this in the abstract may need better justified. Unless the LOH length is a big fraction of the chromosome length, would it be normal to assume that more internal LOH tracts than tracts reaching either end of the chromosome?

In fact, mechanistically, the differences between I-LOH and T-LOH events have been described in the literature. One of our findings is the fact that on average, I-LOH events despite being significantly shorter than T-LOH events (7.4 kb vs 55.3 kb), contribute equally to the overall homozygosity of the genome. Gene conversions during mitotic growth are more frequent than crossovers or BIR events and therefore I-LOH events >> T-LOH events (Symmington, Rothstein and Lisby, 2014). In addition, most terminal LOH and several I-LOH events represent a substantial part of the chromosome length in several cases (please see Supplementary file 3). The importance of these numbers has been described in the results and Discussion sections.

3. In the introduction, they wrote "The diploid organisms (or those with a higher ploidy) are usually heterozygous for many of these genetic variants across the genome". This statement is not sophisticated as many diploid organisms are asexually reproduced or self-fertilized.

This is a generalized statement that intend to reflect the fact that most organisms in nature have heterozygous genomes. Additionally, in organisms such as yeast where asexual reproduction and self-fertilization are common, recent population genomics studies have also demonstrated the prevalence of heterozygosity (Peter et al. 2018; Fischer, Liti and Llorente 2020).

4. In the introduction, they wrote "The mechanism behind LOH have been largely derived" have ->has? (or mechanism ->mechanisms?)

The error has been corrected (mechanism->mechanisms).

5. The introduction section needs more background information on the mechanisms and what is known about I-LOH and T-LOH events. At the same time, the first two paragraphs need to be tightened up and focusing just on the relevant.

Additional text has been incorporated into the introduction to further describe mechanisms behind I-LOH and T-LOH (lines 87-111 and 369-388). In addition, these mechanisms were also discussed in detail in the context of the Results section. We think that the first two paragraphs are relevant and have been intentionally generalized to highlight the role of heterozygosity and LOH in the context of genotype-phenotype diversity and disease development.

6. They wrote "a high LOH level with 25 regions covering approximately 50% of the genome on average was observed in this large panel (11)". Details/specifics need to be provided for the high LOH level and length of the regions. Bring some of the information early.

We refer to the findings of Peter et al. 2018, where whole-genome sequencing of 1011 *S. cerevisiae* natural isolates revealed the extensive levels of LOH in the genome and further encouraged us to pursue the current study. However, the levels and size of LOH regions are not relevant as the methodology in Peter et al. 2018 is heavily biased towards detecting only the large T-LOH. Further, our studies highlight that the short I-LOH events are equally relevant in the context of genome evolution.

7. They wrote "we performed MA experiments by using a set of different *S. cerevisiae* diploid hybrids, designed to mimic the wide genetic diversity observed in this species." Again, details on the hybrid need to be spelled. How many? Rationale of using them. Move up the text from the back.

The text has been updated and the rationale and details have been described in the abstract Lines 41-45, introduction Lines 124-133, results 138-158, Tables 1 and Supplementary file 1.

8. They wrote "The LOH rate is always much higher than the mutation rate"The difference needs to be described with some real information. Slightly higher, several fold or several orders of magnitude? The measured LOH rates in this study really need to be compared with the LOH literature. The authors should consider expand this with some in-depth comparison and discuss the comparison into a big picture.

LOH rates are five orders of magnitude higher than base mutation rates (incorporated in text line 299-300).

9. They wrote "Selection is known to be minimal ". With real data in hand, why assume? They authors should analyze the MA data and perform selection test. The test has been done routinely in many MA studies. The section "Set up and propagation of MA lines" should be moved to the Methods section.

Mutation accumulation experiments were designed to minimize the role of natural selection (Halligan and Keightly 2009; Lynch et al. 2016). Additionally, we find that the SNM rates in our lines are not different from previous estimates and that all established mutational biases exist in our dataset. Therefore, we did not perform the selection test. The “set and propagation of MA lines” section is essential, in our view, to move seamlessly to the LOH section of the results and to make it easier for the readers to have the experimental design in mind. Please also see, reviewer 1 comments 4, 5, 6 and their responses.

10. In results, they wrote "In *S. cerevisiae*, LOH events can be either centromere proximal, interstitial (I-LOH) resulting from gene conversions or centromere distal, terminal (T-LOH) resulting from mitotic crossovers or BIR." More background information like this is needed in the introduction to prepare the readers.

We have incorporated text in the introduction (lines 89-98 and lines 105-111) on the role of DSB repair during mitosis and meiosis as well as meiotic abortions in the generation of LOHs. In addition, we also added text to the discussion (lines 369-372 and lines 378-388).

11. The nearly homozygous lines are interesting. How the nearly homozygous lines differ from each parent along the chromosomes (recombination break points?) Do they result from inbreeding (selfing)?

In fact, 3-7 rounds of intra-tetrad mating following meiosis is required to achieve 70- 90% of genome-wide LOH (Knop 2006; Dutta et al. 2017). The MA lines (all except H9-8 and H9-11) do easily sporulate on SPO media (1% KAc). We never observed sporulation in any of the MA lines at all bottlenecks (25, 50, 75, 100) or the ancestral diploids on YPD. Therefore, we can only speculate on the reasons behind the origin of the nearly homozygous lines to be hyperrecombination, return to growth following abortive meiosis, sib-sib mating or large-scale chromosome loss and reduplication. For the nearly homozygous (NH) lines, fraction of genome under LOH varies from 88 to 98% and the recombination breakpoints are not conserved within genetic backgrounds (Figure 1—figure supplement 5).

12. They wrote "While the I-LOH rates were 3.1 x 10^-5^ per site per division and 5.6 x 10^-2^ events per division, we found that the T-LOH rates were 4.2 x 10^-5^ per site per division and 9.2 x 10^-3^ events per division." The comparison between I-LOH and T-LOH events is potentially interesting. Do the selected hybrids have disproportionately different levels of heterozygosity at the end of the chromosomes? Could the difference in heterozygosity at the end of chromosomes explain the difference between I-LOH and T-LOH among them?

We compared the mean SNP counts for 5 kb windows at the chromosome ends (first/last 20 kb) and the rest of the chromosomes (Author response table 1) in all the ancestral diploids, H1-H9. In fact, we find, the SNP densities to be lower at chromosome ends although not disproportionately different (Wilcoxon p<0.01 for H1-H8 and p>0.05 for H9).

**Author response table 1. resptable1:** Mean event counts for 5 kb at chromosome ends vs rest of the genome.

**Hybrid** **Background**	**Mean density** **at chr_ends**	**Mean density** **rest_genome**
H1	3.66	3.98
H2	7.06	6.81
H3	7.57	9.26
H4	15.00	22.19
H5	20.05	23.67
H6	20.27	22.80
H7	31.54	31.78
H8	36.20	46.73
H9	46.58	47.70

13. They wrote "we have indeed found a positive correlation between the heterozygosity level and the number of T-LOH events." How much of this is due to the fact that many LOH events cannot be effectively detected in low heterozygous sequences.

The average SNP density ranges from 0.8 to 9.7 SNPs per kb. T-LOH events are typically large events and even in our dataset the average T-LOH size is 55.2 kb. Although possible these events are less likely to be overlooked. In addition, by excluding the least heterozygous H1 lines from the analysis, the correlation still remains strong (R=0.64, p<2.2 x 10^-16^).

Reviewer #3In this study, the authors set out to analyze Loss of heterozygosity in 160 MA lines derived from 9 diploid ancestors (mating of 2 haploid parents). Main results include: LOH is higher than point mutations, interstitial LOH was more frequent than terminal LOH, the proportion of LOH, LOH spectrum and rate of LOH differed and was dependent on the linage (genetic background), terminal LOH events were more frequent in ancestors with higher LOH levels, and short interstitial LOH events were more frequent in ancestors with low fertility.1. The result section seems to be a bit unfocused and quite wordy/lack of clarity. I would suggest editing and shorten for a more concise description of the results.

Based on comments from all reviewers we have incorporated changes to the introduction, Results section, and Discussion sections for better reading.

2. For the discussion, the authors exclusively focus on the literature on Saccharomyces species. There is a lot of recent data on LOH for other yeast species (Candida) that I strongly recommend the authors discuss. In addition, there are a few spots that should probably go into the discussion. For example, comparisons to previous studies.

We agree, several organisms have been central to the growing understanding of LOH in genome evolution. We have now introduced and discussed LOH in the context of other species as well (e.g., *D. pulex* (Flynn et al. 2017), *Cryptococcus sp*. (Li et al. 2012) and *Candida albicans* (Forsche et al. 2011, 2018; Ropars et al. 2018, Ene et al. 2018)). Unfortunately, the literature referring to the scope our study was limited to *S. cerevisiae*. Similar mechanisms can certainly be at play in other organism as well, we refrained from drawing these parallels. In addition, LOH rate variation in *Candida albicans* have been reported in response to environmental changes (Bouchonville et al. 2009; Li et al. 2012; Forche et al. 2018). Please also see, response to Reviewer 2 comment 1 and 8.

3. The first result paragraph contains mostly methods and should probably be moved there. It would be helpful to maybe add a figure with a flowchart for the MA propagation.

We think it is important that this paragraph is present at the beginning of the results because it gives the reader more clarity on the design of the study and allows a better understanding of the results afterwards.

4. Was the genomic DNA for WGS extracted from single colonies?

DNA extractions were performed from the single colonies isolated at the end of the experiment.

5. The cut-off of two SNPs to call it LOH is quite low. What is the percentage of these 2-3 SNP LOH events compared to the total?

The distribution of the number tracts supported by two to >=20 SNPs has been shown in Author response image 1. We used a low two SNP cut off since there was a 12-fold difference (H1 vs H9) between the SNP densities across the hybrids. Higher cut-offs would not allow us to detect short events in the backgrounds with low SNP densities. In addition, our results also hold true at 3 SNP and 4 SNP cut offs as well. These short LOH tracts have been a very important part of the LOH spectrum.

**Author response image 1. sa2fig1:** 

6. Have the authors confirmed at least a few of the LOH events by Sanger sequencing? This could be done easily on short LOH (<500 bp).

The final dataset of LOH events were generated by a stringent QC during the bioinformatics analysis. Moreover, both the large and short tracts were also detected in lines with very high coverage (>100X). Therefore, we opted not to sanger verify the LOH tracts.

7. For the lines with almost complete LOH, did the authors find evidence for aneuploidy, for example regions with fewer/more reads?

We now have addressed the aneuploidy status in the MA lines in the main text (lines 321-328). Overall, we found no difference in aneuploidy rates in the nearly homozygous lines compared to the rest.

8. There are multiple color schemes within figures, which makes it harder to focus. Would it be possible to maybe have the yellow/blue (e.g., Figure 2B) and/or the blue/red (violin plots) in gray scale?

Color schemes have been changed as suggested in all figures. Only the yellow and blue color scheme has been retained to distinguish between interstitial and terminal LOH events.

9. Can the authors please comment on why it was important to check for spore viability? From Figure 3C it looks like the spore viability correlates with parental fertility and therefore was dependent on their genotype?

In nature, hybridization events causing extensive heterozygosity lead organisms to exploit new environments and speciation in the long run. Although, *S. cerevisiae* hybrids are frequently viable, intra-specific meiotic infertility is often observed (Xu and He 2011; Hou et al. 2016). Therefore, for facultatively sexual organisms, meiotic fertility is essential (Morales and Dujon 2012). Mitotic or meiotic (inbreeding) LOH is known to be one of the primary modes of fertility recovery in yeast. We were curious to observe the impact of the variable levels of LOH on fertility recovery. As the results suggest, even relatively lower levels of LOH can show a gradual fertility rescue (Charron et al. 2019). The MA lines were grouped as high fertility and low fertility based on their parental fertility state. Ancestral hybrids with meiotic spore viability >75% and fraction of four-spore viable tetrads >50% were designated as high fertility. The ancestors with high fertility on average displayed a spore viability of 86% and fraction of four-spore viable tetrads of 75%, while for the low fertility ancestors group these values were 46% and 36% respectively. The meiotic spore viability did not significantly change in the high fertility group. However, the viability significantly increases in the MA lines from the low fertility group compared to the ancestral viability, as a consequence of the LOH.

10. For Figure 1—figure supplement 1, the color choice of black and grey makes it hard to visually identify SNPs versus homozygosity. I recommend changing that.

Figure 1—figure supplement 1 has been modified as suggested.

11. Looking at figure 1—figure supplement 2, there seems to be no correlation of LOH and chromosome size. And while the graphs in figure 2—figure supplement 1 show a slight correlation, none of them are significant. Therefore, I am not sure if figure 2—figure supplement 1is needed. Similarly, I don't think Figure 2—figure supplement 2 is necessary. How do the authors define low and high fertility and what is the cut-off?

Figure 1—figure supplement 2 represents the distribution of the fraction of the genome under LOH and may not be the best way to draw this conclusion. However, Figure 2—figure supplement 1 shows a very strong correlation between I-LOH and chromosome size but not in T-LOH. Figure 2—figure supplement 2, we agree, was not particularly useful and has been replaced.

12. Please define MA at first use.

MA was defined at first use in the introduction (Line 125).

13. Please add (SC) when describing the synthetic complete media recipe.

SC has been added while describing synthetic complete media.

14. In Figures 1A and 1B, 2A-C, 3A, figure 2—figure supplement 1, figure 2—figure supplement 2, figure 2—figure supplement 3-figure 4—figure supplement 3, I suggest adding the term LOH to the y-axis titles. The x-axis title font in figure 3 needs to be a bit larger. Also, it is impossible to read the lineage ID on the circles in 3C. Maybe a color legend would be better here.

Legends have been changed as suggested across all figures.

15. Please define NH and SNM in the figure legends (e.g., Figure 1—figure supplement 4, Figure 4—figure supplement 1).

NH and SNM have been defined in the legends as suggested across all figures (Lines 796-801).

16. In figure 1—figure supplement 5, please indicate the centromere in a different color.

The centromeres have now been indicated by vertical dashed lines in Figure 1—figure supplement 5.